# Coupled Effects of Stress, Moisture Content and Gas Pressure on the Permeability Evolution of Coal Samples: A Case Study of the Coking Coal Resourced from Tunlan Coalmine

**Guofu Li** [1,†]**, Yi Wang** [2,*]**, Junhui Wang** [2,3,*]  **, Hongwei Zhang** [1,4,5,*,†]**, Wenbin Shen** [2] **and Han Jiang** [4,5]

1   State Key Laboratory of Coal and CBM Co-Mining, Jincheng 048000, China; 13834068216@163.com
2   College of Safety and Emergency Management Engineering, Taiyuan University of Technology, Taiyuan 030024, China; akaite521@126.com
3   Key Laboratory of Deep Coal Resource Mining (CUMT), Ministry of Education of China, School of Mines, China University of Mining and Technology, Xuzhou 221116, China
4   Beijing Key Laboratory for Precise Mining of Intergrown Energy and Resources, China University of Mining and Technology (Beijing), Beijing 100083, China; jianghan722@163.com
5   School of Energy and Mining Engineering, China University of Mining and Technology (Beijing), Beijing 100083, China
*   Correspondence: wangyi@tyut.edu.cn (Y.W.); wangjunhui@cumt.edu.cn (J.W.); hongwei@cumtb.edu.cn (H.Z.)
†   These authors contributed equally to this work.

**Abstract:** Deep coalbed methane (CBM) is widely distributed in China and is mainly commercially exploited in the Qinshui basin. The in situ stress and moisture content are key factors affecting the permeability of $CH_4$-containing coal samples. Therefore, considering the coupled effects of compressing and infiltrating on the gas permeability of coal could be more accurate to reveal the $CH_4$ gas seepage characteristics in CBM reservoirs. In this study, coal samples sourced from Tunlan coalmine were employed to conduct the triaxial loading and gas seepage tests. Several findings were concluded: (1) In this triaxial test, the effect of confining stress on the permeability of gas-containing coal samples is greater than that of axial stress. (2) The permeability versus gas pressure curve of coal presents a 'V' shape evolution trend, in which the minimum gas permeability was obtained at a gas pressure of 1.1MPa. (3) The gas permeability of coal samples decreased exponentially with increasing moisture content. Specifically, as the moisture content increasing from 0.18% to 3.15%, the gas permeability decreased by about 70%. These results are expected to provide a foundation for the efficient exploitation of CBM in Qinshui basin.

**Keywords:** coal bed methane; permeability; in-situ stress; moisture content

## 1. Introduction

Coalbed methane (CBM) has become one of the most promising potential unconventional gas sources in the coal mining field globally [1,2]. This resource is widely distributed in China, and the total amount of CBM resources buried shallower than 2000 m is $36.8 \times 10^{12}$ m³ [3]. With respect to CBM extraction, the permeability evolution of gas-containing coal under the in-situ stress should be fully understood before commencing a commercial CBM project. The permeability of coal is influenced by many factors, such as in situ stress, moisture content, gas adsorption and loading/unloading conditions. Normally, the porosity of coal decreases with increasing buried depth, resulting in the low permeability of CBM reservoirs, thus affecting the effective extraction of CBM. Previous studies have indicated that the permeability of CBM reservoirs always presents an exponential decay trend with increasing burial depth [4]. However, some studies showed that coal permeability does not always decrease with the increase of stress, because after the stress increases to a certain degree, the coal body enters the elasto-plastic stage, which results in

the generation of new micro-cracks and enhancing its permeability [5]. Therefore, further understanding the influence of stress on the permeability of coal containing water and gas is significant to the exploitation of CBM.

Coal permeability is highly sensitive to stress. To date, many studies have revealed the relationship between effective stress and the permeability of gas-containing coal. For example, Somerton et al. [5] conducted a permeability test of coal samples filled with $CH_4$ gas. Pan et al. [6] found that the permeability declined with increasing pore pressure at a constant effective stress. Yin et al. [7,8] established the relationship between permeability and the effective stress of coal under loading/unloading conditions. The results indicated that the permeability of gas-containing coal decreases with the increase of axial stress and confining stress. Conversely, the permeability increases with increasing gas pressure. Yuan et al. [9] pointed out that the permeability first decreases and then increases with increasing $CH_4$ pressure in coal during loading. A turning point was obtained at a gas pressure of around 1.2 MPa. Similarly, Cao et al. [10] found a 'V'-shaped permeability versus gas pressure curve for outburst-prone coal. Li et al. [11] studied the changes in porosity and permeability of coal samples under various confining pressures. They found that the porosity drops faster than permeability with the increase of confining stress. By filling the coal with Helium gas to measure its permeability, Li et al. [12] found that the coal permeability first decreases and then increases with decreasing pore pressure. Furthermore, when the pore pressure is less than 1.9 MPa, the effective stress and gas slippage effects simultaneously control the coal permeability. Meanwhile, when the pore pressure is greater than 1.9 MPa, the effective stress plays a major role in controlling the coal permeability.

Naturally, water and vapor are adsorbed in the coal matrix. The CBM reservoirs are soaked up by water during drilling and drainage. Therefore, considering the influence of moisture content on the permeability of gas-containing coal can objectively reflect the in situ state and flow characteristics of coal during CBM exploitation. However, most experiments have been conducted using dry coal samples, and few have been carried out considering moisture contents. Wei et al. [13] found that the permeability of gas-containing coal decreases in a negative exponential manner with increasing moisture content. Jing et al. [14] also found this negative exponential decreasing trend for coal permeability. In addition, they indicated that the permeability decreased faster with increasing moisture content (below 6%), while when the moisture content was higher than 6%, the influence of moisture content on permeability was weakened. Moisture content can affect the gas adsorption capacity within the range of critical content, while beyond this content, the additional moisture is not able to influence gas adsorption. The adsorption capacity of methane can increase by 35–40% if it is stored in a humid environment for one year [15]. Generally, the influence of moisture content on the gas adsorption capacity depends significantly on the rank of the coal. Specifically, low-rank coal is affected by moisture sorption more significantly than high-rank coal [16].

As mentioned above, a lot of studies have investigated the factors influencing the permeability of gas-containing coal. Furthermore, it is confirmed that the relative permeability and reservoir pressure change with the increasing impact radius around an injection well in typical hydraulic fracturing engineering [17,18]. Moisture always coexists with gas in coal at all five stages of gas recovery around an injection well. The fluids-gas flow of coal in an in situ CBM extraction project is very complex to understand and is influenced by factors including stress, gas pressure, moisture, cleats-pore system, temperature, etc. All these factors influence the flow behavior of gas in coal. However, this study focuses more on three factors (i.e., moisture, gas pressure and stress), which would be three basic variables for further understanding the in situ flow behavior of coal seams. In particular, the effect of individual vertical and horizontal stress on the permeability of gas-containing coal seams needs to be further investigated. To this end, we introduce a theoretical model and a coal permeability testing method. Subsequently, the coupled effects of axial stress, confining stress, moisture content and gas pressure on the permeability of coal are presented. Finally, some conclusions are drawn.

## 2. Theoretical Foundations

$CH_4$ gas mainly migrates by diffusion in the coal matrix, while it migrates by seepage in the pores and fractures of coal. The speed of gas migration by seepage is much greater than that by diffusion. Generally, gas flow in coal pores and fractures obeys Darcy's law, and the permeability model can be expressed as:

$$K = \frac{2QP_0\mu l}{(P_1{}^2 - P_2{}^2)A}$$ (1)

where $K$ is the permeability of gas, $10^{-3}$ μm$^2$; $P_0$ is the standard atmospheric pressure, $10^5$ Pa; $Q$ is flow flux, cm$^3$/s; $\mu$ is the dynamic viscosity of $CH_4$ gas, which is determined as $1.08 \times 10^{-5}$ Pa·s at room temperature; $P_1$ and $P_2$ are inlet and outlet gas pressures, respectively, Pa; $l$ is the length of the sample, cm; $A$ is the cross-sectional area of sample, cm$^2$.

Effective stress is a key factor in deforming coal samples and determining gas permeability. Normally, effective stress is the difference between the external ground pressure and the internal gas pressure of coal. The equation can be expressed as follows:

$$\sigma_0 = \frac{1}{3}(\sigma_v + 2\sigma_h) - \frac{1}{2}(P_1 + P_2)$$ (2)

where $\sigma_0$, $\sigma_v$ and $\sigma_h$ are effective stress, axial stress and confining stress, respectively, MPa; $P_1$ and $P_2$ are inlet and outlet pressures of gas, respectively, MPa.

## 3. Methodology

### 3.1. Sample Preparation

Coal is an extremely complex porous organic rock in which lots of cleats and pores are developed. According to the dual-porosity coal structure model, the inner space of the coal body consists of pores in the coal matrices and fractures around the coal matrices. In this study, we focused more on the gas flow in the pores.

Coal samples were sourced from the Tunlan coalmine near Qinshui Basin, Shanxi, China (Figure 1). The absolute gas emission of this coalmine is about 260 m$^3$/min, and the NO. 8 and NO. 9 coal seams are closely overlapped with a total thickness of about 10 m and with a vertical spacing distance between 0.1 and 2.4 m. The employed coal blocks were taken from the NO. 8 coal seam at a buried depth of more than 800 m. The sourced coal blocks were processed into ISRM suggested samples with a dimension of $\varphi$50 mm × 100 mm (Figure 2a). It should be noted that we tried to drill intact coal without cleats during the sampling procedures. Samples were rejected if obvious cleats were found or if a cleat penetrated the coal sample. Finally, 40 samples were made and tested in this study. The basic properties of the coal sourced from the NO. 8 coal seam of the Tunlan coalmine are listed in Table 1 in detail.

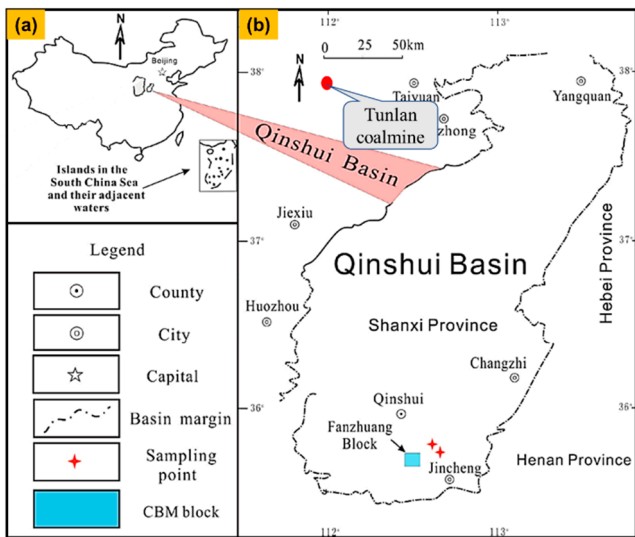

**Figure 1.** Maps of the Qinshui Basin (**a**) and the Tunlan coalmine (**b**) (after Zhao et al. [19]).

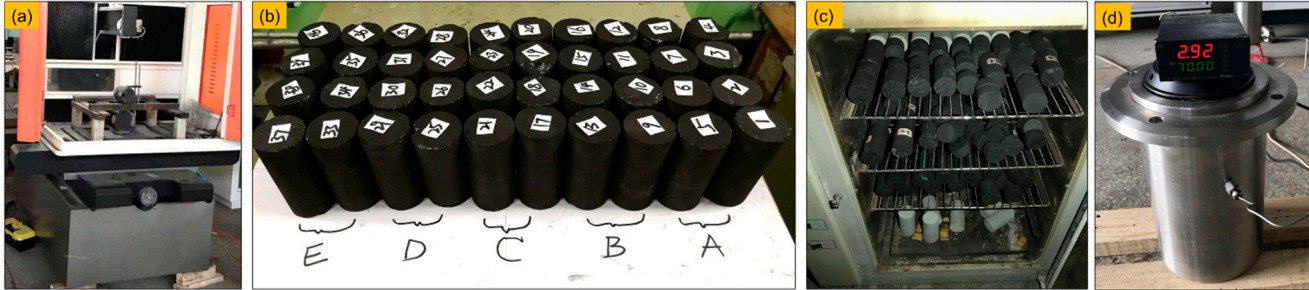

**Figure 2.** Sample preparation setups. (**a**) Line cutting machine; (**b**) coal samples; (**c**) drying oven; (**d**) water injection apparatus.

The initial moisture content of the Tunlan coal sample was determined to be 1.61%. In order to prepare samples with different moisture contents, the coal samples were dried using a heating control system at a constant temperature of 105 °C to achieve moisture contents of 0.18% and 0.76%. Afterward, the original coal sample with a moisture content of 1.61% was connected to a manual pressure test pump and a closed kettle in order to inject water at a constant high pressure of 5 MPa into the coal sample. After maintaining the injection pressure for 24 h, coal samples with a moisture content of 3.15% were obtained. Finally, four groups of coal samples with different moisture contents (i.e., 0.18%, 0.76%, 1.61% and 3.15%) were obtained (Figure 2b).

**Table 1.** Basic properties of Tunlan NO. 8 coking coal.

| Properties | Detail | Unite | Value |
|---|---|---|---|
| Physical | Density, $\rho$ | g/cm$^3$ | 1.32 |
| | Porosity, $\varphi$ | % | 2.86 |
| Mechanical | Tension strength, $\tau$ | MPa | 1.29 |
| | Uniaxial compression strength, $\sigma_c$ | MPa | 10.1 |
| | Elastic modulus, $E$ | GPa | 5.1 |
| | Poisson's ratio, $v$ | N/A | 0.33 |
| | Cohesive strength, $c$ | MPa | 1.83 |
| | Internal friction angle, $\varphi_o$ | ° | 35 |
| Proximate analysis | Air-dried moisture, $M_{ad}$ | % | 1.61 |
| | Dry base ash, $A_{ad}$ | % | 6.86 |
| | Dry ash-free volatiles, $V_{daf}$ | % | 26.09 |
| Gas parameters | Gas content | $m^3/t$ | 8.631–15.49 |
| | Absorption constant, $a$ | cm$^3$/g$_{daf}$ | 23.12 |
| | Absorption constant, $b$ | MPa$^{-1}$ | 1.08 |

### 3.2. Experimental Apparatus and Methods

The experiments were conducted by employing the WYS-800 triaxial loading–gas seepage test setups. This apparatus consists of a loading frame, a servo-hydraulic station, an air path system, a triaxial vessel, a constant-temperature oil bath, and a data acquisition system. The schematic diagram of the testing apparatus is displayed in Figure 3. The main technical parameters are as follows: axial stress range 10–800 KN, confining stress range 0–15 MPa and gas pressure range 0–15 MPa.

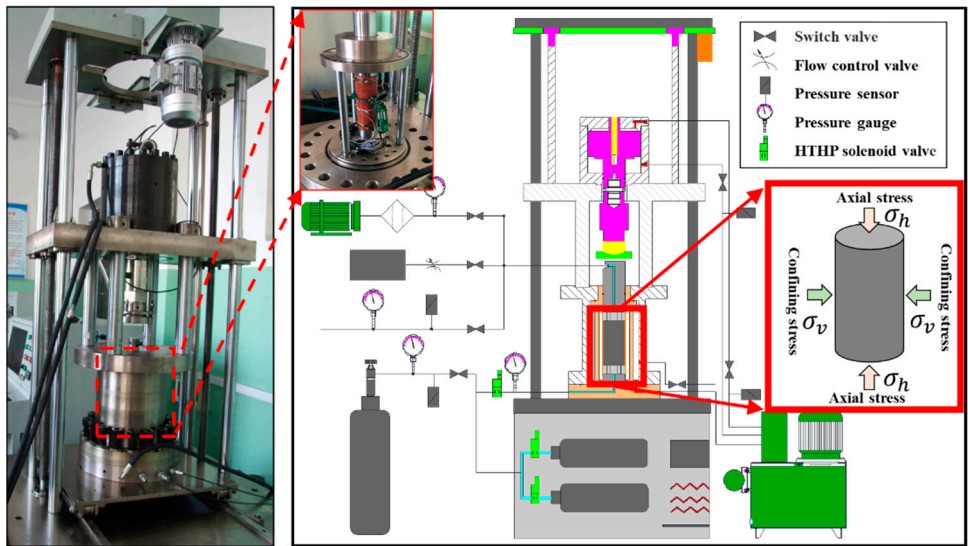

**Figure 3.** Picture and structure diagram of test apparatus.

The test methods are described as follows: (1) Fully wrap the coal sample with a Teflon jacket and then locate the sample inside the triaxial vessel. (2) Load the confining stress to 1 MPa at a rate of 0.01 MPa/s and apply the axial stress of 5 KN at a speed of 0.1 KN/s. After the initial stresses become stable, load the axial and confining stresses alternately until the target stresses are achieved. (3) Inject high-purity CH$_4$ gas into the

coal sample. After this has been fully absorbed and desorbed on the coal sample, remove the residual air in pipes and open the outlet and inlet flow valves. Then, measure the gas permeability. (4) Close the outlet flow valve and finish a test. (5) Repeat step 1 to step 4 for the remaining coal samples to complete all tests. Overall, the confining stress in these tests varies from 2 to 8 MPa, and the gas pressure varies from 0.5 to 1.4 MPa.

## 4. Results and Discussion

### 4.1. Effect of Axial Stress on the Gas Permeability of Coal Samples

Originally, the coal is reserved in a balanced stress state in the depths. Following underground coal mining, this initial balanced stress state is broken. Consequently, the vertical and horizontal stresses of coal seams are changed, resulting in evolutions in the pore structure and permeability of the coal mass. Since the axial/normal stress and lateral/confining stress have different effects on the permeability of coal samples, in this section, we first present the influence of normal stress on the permeability evolution of gas-containing coal at two moisture contents (0.18% and 3.15%) and under a given confining stress (4 MPa). It should be noted that the axial stress during the sample loading process is gradually increased by a stress rate of 1 MPa per step until the coal sample fails. The permeability is measured intermittently during axial stress loading. As shown in Figure 4a, the permeability versus normal stress curves of gas-containing coal samples can be roughly described by a 'V' shape, in which the gas-containing coal permeability first decreases and then increases with increasing axial stress.

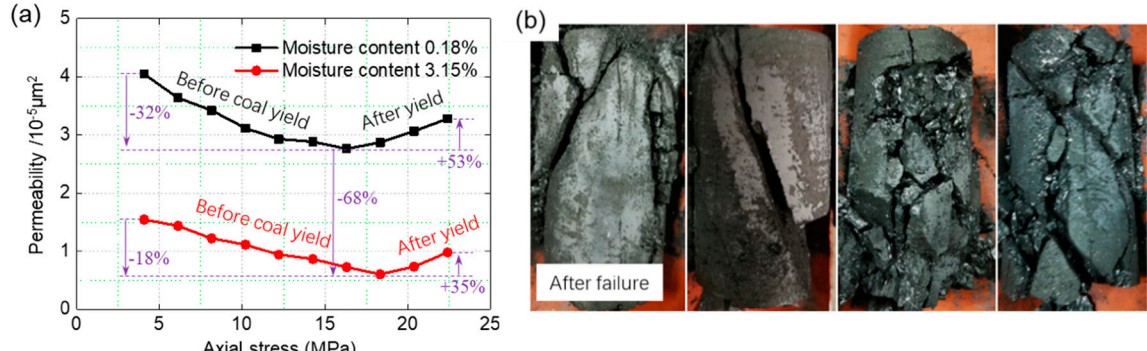

**Figure 4.** Gas permeability versus axial stress curves of coal (**a**) and coal samples after failure (**b**).

During the triaxial compression of coal samples, the obtained stress-strain curve can be roughly divided into two stages, i.e., pre-yield and post-yield stages. In the pre-yield stage, the coal sample is first compacted and then undergoes elastic deformation. In the first compaction process, an upwards concave trend in the stress–strain curve can be observed, which may be due to the closure of primary cracks, pores and voids in coal samples [20]. During the elastic deformation process, the loading increases linearly with axial strain. Following the elastic deformation, the yield strengths are exceeded, after which it enters into the post-yield strength stage. Subsequently, the plastic deformation and macroscopic failure of the coal sample will occur in the post-yield stage.

The deforming and failure response is closely related to the evolution of gas permeability in coal samples. During the first compaction process before coal yield, the internal cracks, pores and voids are gradually compacted, resulting in the gas migration channel closing and permeability decreasing. Furthermore, the internal pore structure of the coal sample is further compacted during the elastic deformation of coal. Therefore, the permeability drops continuously to a minimum value during the pre-yield stage. The minimum gas permeabilities of coal samples with moisture contents of 0.18% and 3.15% are about $2.75 \times 10^{-5}$ $\mu m^2$ and $0.6 \times 10^{-5}$ $\mu m^2$, respectively.

The increase in permeability mainly occurs after the coal sample yields. During this stage, the internal cracks in the coal are initialized and developed. The gas migration

channels in the coal are gradually increased and connected, thus resulting in enhanced permeability. With a further increase in axial stress, the coal body enters a macroscopic failure process (Figure 4b), greatly improving gas permeability.

### 4.2. Effect of Confining Stress on the Gas Permeability of Coal Samples

This section presents the evolution of the gas permeability of coal samples under the variable confining stresses. In this test, the axial stress and gas pressure were set separately as 10 MPa and 0.8 MPa. The moisture contents were set as 0.18%, 0.76%, 1.61% and 3.15%. As shown in Figure 5, the gas permeability of coal decreases sharply with increasing confining stress. This is because the internal defects of the coal body are gradually compacted with increasing confining stress, leading to the closing of gas migration channels, thereby reducing the gas permeability of the coal.

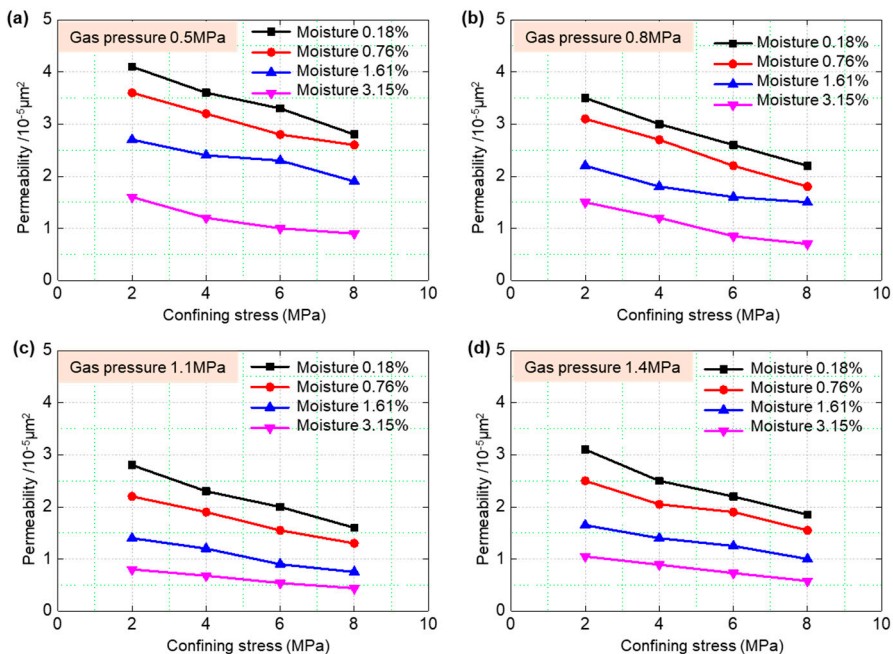

**Figure 5.** Gas permeability–confining stress curves of coal with viable moisture contents (0.18%–3.15%) and gas pressures (0.5MPa (**a**), 0.8MPa (**b**), 1.1MPa (**c**) and 1.4MPa (**d**)).

The decreasing trend in permeability gradually slows down. That is to say, the permeability greatly decreases in the relatively low confining stress stage, and it decreases slightly in the high confining stress stage. When the confining stress is low, there are few contact points between the matrix and the mineral particles in the coal. Thus, the pores are weakened in their ability to resist deformation, and the pore size is greatly compressed. As the stress increases, the pores are further compacted and closed, but the contact points between the particles increase. Therefore, the ability of pores to resist deformation is enhanced, and pore size changes slightly.

### 4.3. Coupled Effects of Axial and Confining Stresses on the Gas Permeability of Coal Samples

Figure 6a shows the evolution of the gas permeability of coal with increasing axial stress (within the elastic stage of coal) under confining stresses of 6 MPa and 8 MPa. Similarly, Figure 6b presents the changes in coal permeability with increasing confining stress under axial stresses of 8 MPa and 10 MPa. The power fitted curves obtained using the equation $K = ae^{b\sigma}$ are shown in Figure 6 (where, $K$, and $\sigma$ are gas permeability and stress, respectively; $a$ and $b$ are the fitted parameters). Correspondingly, the fitted equations are presented in Table 2 in detail.

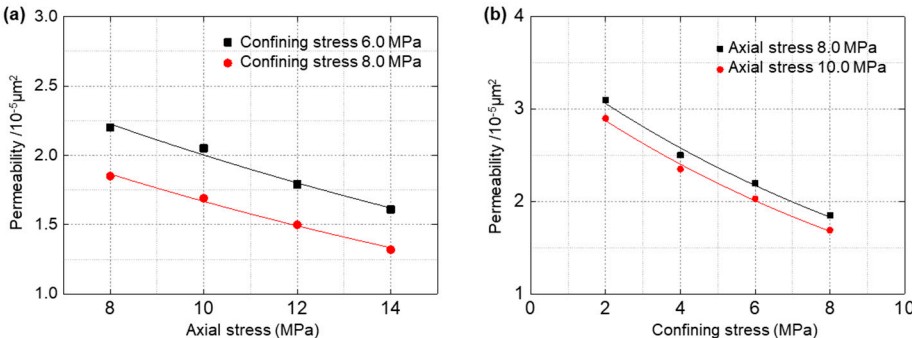

**Figure 6.** Coupled effects of axial stress (**a**) and confining stress (**b**) on the gas permeability of coal.

**Table 2.** Fitted details of the gas permeability of coal under various loading conditions.

| NO. | Axial Stress $\sigma_v$/MPa | Confining Stress $\sigma_h$/MPa | Fitting Equation $K=ae^{b\sigma}$ | $R^2$ |
|---|---|---|---|---|
| 1 | 2, 4, 6, 8 | 6 | $K = 2.39567e^{-0.05282\sigma_v}$ | 0.97712 |
| 2 | 2, 4, 6, 8 | 8 | $K = 2.01317e^{-0.05575\sigma_v}$ | 0.99075 |
| 3 | 8 | 8, 10, 12, 14 | $K = 3.63198e^{-0.08550\sigma_h}$ | 0.98421 |
| 4 | 10 | 8, 10, 12, 14 | $K = 3.43816e^{-0.08948\sigma_h}$ | 0.99241 |

As expected, the correlation coefficients $R^2$ fitted by the exponential function are all higher than 0.977. The average values of the fitting parameter $a$ for the permeability–axial stress and permeability–confining stress curves were 2.20442 and 3.53507, respectively. The average values of the fitting parameter $b$ were $-0.054285$ and $-0.08749$, respectively. Therefore, it can be concluded that the influence of confining stress on the permeability of gas-containing coal samples is greater than that of axial stress.

During triaxial compression and gas permeability tests, the gas flow path is consistent with the axial loading direction. The axial loading generally has two effects on the evolution of the pore and fissure structures of coal. One effect is to gradually close the pores and cracks, thus weakening the permeability of the coal. Another is that the coal body will produce a certain lateral expansion during loading, which helps open the gas flow channel and enhance the permeability. Therefore, the second effect results in the decreasing trend of gas permeability. With respect to the confining effect of coal samples, the applied confining stress directly closes the cracks and pore structure from the lateral direction, thus weakening the gas permeability of the coal samples. Additionally, the confining stress limits the lateral expansion effect produced by axial loading, thus further repressing gas flow. For these reasons, confining stress has a greater influence on the gas permeability of coal than axial stress.

### 4.4. Effect of Gas Pressure on the Gas Permeability of Coal Samples

The gas permeability versus gas pressure curves of coal samples under confining stresses of 2, 4, 6, and 8 MPa are shown in Figure 7. There is a nonlinear relationship between the gas permeability of the coal samples and gas pressure. The gas permeability presents a decreasing trend an increase in coal gas pressure from 0.5 MPa to 1.1 MPa, reaching its minimum value at a gas pressure of 1.1 MPa. Under confining stresses of 2, 4, 6 and 8 MPa, the gas permeabilities of coal at a gas pressure of 1.1 MPa decrease by 48%, 50%, 61%, and 61%, respectively. Conversely, when the gas pressure in coal is greater than 1.1 MPa, the permeability versus gas pressure curve shows an increasing trend. Furthermore, under the conditions of 2, 4, 6 and 8 MPa, the gas permeability of coal at a gas pressure of 1.4 MPa compared to a gas pressure of 1.1 MPa is increased by $0.25 \times 10^{-5}$ µm², $0.2 \times 10^{-5}$ µm², $0.35 \times 10^{-5}$ µm² and $0.25 \times 10^{-5}$ µm², respectively. Therefore, an obvious inflection point at a gas pressure of 1.1 MPa can be concluded.

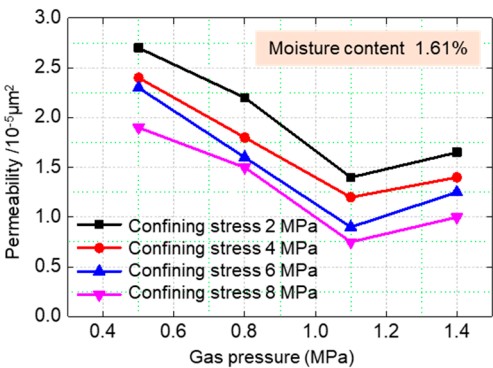

**Figure 7.** Gas permeability versus gas pressure curves of coal samples.

With increasing gas pressure in coal, the amount of gas adsorption gradually increases, and the gas slippage effect or Klinkenberg effect causes the gas permeability to gradually decrease. However, with a continued increase in gas pressure, the slippage effect is gradually released, and the influence of the gas pressure becomes the control factor. Therefore, the main control factor changes to gas pressure with increasing gas pressure, thus enhancing gas permeability.

### 4.5. Effect of Moisture Content on the Gas Permeability of Coal Samples

Figure 8 presents the permeability–moisture content curves under different gas pressures (0.5, 0.8, 1.1, and 1.4 MPa) and confining stresses (2, 4, 6 and 8 MPa). The fitting curves are correspondingly presented in Figure 8, and the fitting details are shown in Table 3 in detail. As shown in Figure 8, the moisture content greatly influences the gas permeability of coal. Overall, the permeability of the gas-containing coal sample shows a negative exponential decreasing trend with increasing moisture content.

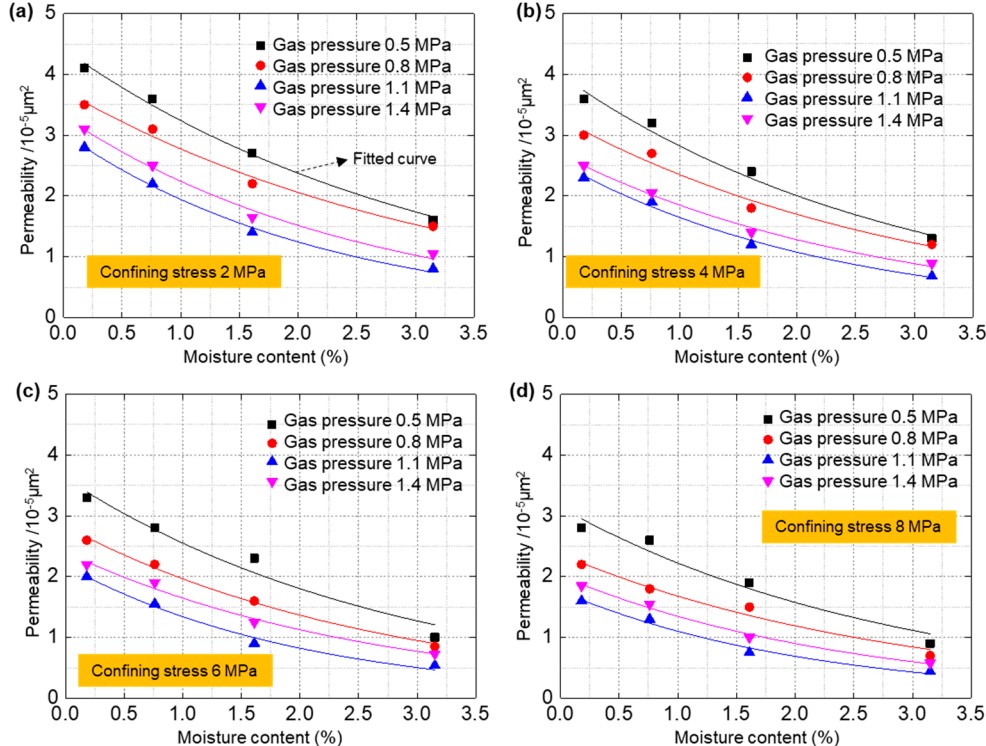

**Figure 8.** Effect of moisture content on the permeability of compressed gas-containing coal. (**a**) confining stress 2 MPa; (**b**) confining stress 4 MPa; (**c**) confining stress 6 MPa; (**d**) confining stress 8 MPa.

**Table 3.** The fitting details of permeability versus moisture content curves of coal samples.

| NO. | Confining Stress/MPa | CH4 Pressure/MPa | Fitting Equation $K=ae^{bw}$ | $R^2$ |
|---|---|---|---|---|
| 1 | 2 | 0.5 | $K = 4.42076e^{-0.31086w}$ | 0.99069 |
| 2 | 2 | 0.8 | $K = 3.73810e^{-0.29813w}$ | 0.98118 |
| 3 | 2 | 1.1 | $K = 3.03553e^{-0.44614w}$ | 0.99296 |
| 4 | 2 | 1.4 | $K = 3.31513e^{-0.39121w}$ | 0.98668 |
| 5 | 4 | 0.5 | $K = 3.96484e^{-0.34131w}$ | 0.96775 |
| 6 | 4 | 0.8 | $K = 3.25065e^{-0.32497w}$ | 0.96543 |
| 7 | 4 | 1.1 | $K = 2.51570e^{-0.42524w}$ | 0.98817 |
| 8 | 4 | 1.4 | $K = 2.66752e^{-0.36704w}$ | 0.9905 |
| 9 | 6 | 0.5 | $K = 3.61192e^{-0.34790w}$ | 0.94446 |
| 10 | 6 | 0.8 | $K = 2.82434e^{-0.36339w}$ | 0.99271 |
| 11 | 6 | 1.1 | $K = 2.18501e^{-0.48835w}$ | 0.98134 |
| 12 | 6 | 1.4 | $K = 2.40452e^{-0.37905w}$ | 0.98305 |
| 13 | 8 | 0.5 | $K = 3.14023e^{-0.34455w}$ | 0.93777 |
| 14 | 8 | 0.8 | $K = 2.36980e^{-0.34400w}$ | 0.96184 |
| 15 | 8 | 1.1 | $K = 1.76251e^{-0.46914w}$ | 0.97761 |
| 16 | 8 | 1.4 | $K = 2.01755e^{-0.40367w}$ | 0.98748 |

As shown in Figure 4a, the moisture content barely changes the normal stress-permeability curve (i.e., 'V' shape) trend of gas-containing coal samples. However, it will greatly affect the coal permeability values. In this case study, the permeability dropped by an average of 68% with an increase in moisture content from 0.18% to 3.15%. Therefore, the change in moisture content does not essentially change the initial structure of the coal samples, but the effective porosity of coal body is occupied by water and/or vapor, thus weakening the gas flow and sharply decreasing the permeability of the coal.

For coal samples with a gas pressure of 0.5 MPa under a confining pressure of 6 MPa, the coal permeability decreased from $3.348543 \times 10^{-5}$ $\mu m^2$ to $1.039368 \times 10^{-5}$ $\mu m^2$ with an increase in moisture content from 0.18% to 3.15%, which represents a decrease of 69%. Therefore, it can be easily concluded that the permeability of gas-containing coal can be significantly limited by increasing its moisture content. This is because the drilled coal is relatively hydrophilic, and water/vapor is adsorbed on the surface of coal particles, thus occupying the internal pore/fracture structures of coal body.

The fitting function for the permeability versus moisture content curves in Figure 8 can be expressed as follows:

$$K = ae^{bw} \tag{3}$$

where $K$ is the coal permeability, $10^{-5}$ $\mu m^2$; $a$ and $b$ are fitting parameters, $w$ is the moisture content of coal.

As expected, the fitted curves agree well with the experimental data, in which all correlation coefficients are higher than 0.93777. Therefore, in the process of CBM extraction or $CH_4$ gas drainage, the permeability can be evaluated by the proposed fitting equations. These results could help predict gas emission amounts and for developing reasonable gas extraction plans.

## 5. Conclusions

Understanding the permeability evolution of gas-containing coal is significant to both the extraction of CBM and the prevention of $CH_4$ gas outburst disasters. In this study, the WYS-800 triaxial gas seepage apparatus was employed to investigate the evolution of gas permeability in the Tunlan coal samples under various stresses, moisture contents and gas pressures. The following conclusions can be drawn:

(1) The permeability of coal samples first decreases and then increases with increasing axial stress, which corresponds to the compressed sealing and failing effects of coal samples under the triaxial compression test. The permeability of coal samples always decreases with increasing confining stress in a negative exponential manner. Furthermore, in our triaxial test, the inhibition impact of confining stress on the permeability of coal samples was greater than that of axial stress.

(2) Similarly, the permeability of coal samples first decreases and then increases with increasing gas pressure, which may be a result of the Klinkenberg effect. The lowest permeabilities of Tunlan coal samples were observed at a gas pressure of 1.1 MPa.

(3) Water or vapor could be adsorbed onto the surface of the coal particles and occupy the internal pore/fracture structures of the coal body, leading to a negative exponential decreasing trend in permeability. For the employed Tunlan coal samples, the coal permeability decreased by about 70% with an increase in moisture content from 0.18% to 3.15%.

**Author Contributions:** Conceptualization, Y.W.; methodology, W.S.; investigation, H.Z.; resources, Y.W. and W.S.; data curation, W.S. and H.J.; writing—original draft preparation, G.L., H.Z. and J.W.; supervision, G.L.; funding acquisition, G.L. and H.Z. All authors have read and agreed to the published version of the manuscript.

**Funding:** This research was funded by The Open Research Fund Program of State Key Laboratory of Coal and CBM Co-Mining, grant number 2019KF04, and Fundamental Research Funds for the Central Universities, grant number 2020XJNY03.

**Institutional Review Board Statement:** Not applicable.

**Informed Consent Statement:** Not applicable.

**Data Availability Statement:** The data used to support the findings of this study are available from the corresponding author upon request.

**Conflicts of Interest:** The authors declare no conflict of interest.

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
