# Peer review of "Coupled Effects of Stress, Moisture Content and Gas Pressure on the Permeability Evolution of Coal Samples: A Case Study of the Coking Coal Resourced from Tunlan Coalmine"

_water, doi:10.3390/w13121653_

Round 1

Reviewer 1 Report

I am submitting my review of manuscript "Coupled Effects of stress, moisture content and gas pressure on the permeability evolution of coal samples: A case study of the coking coal resourced from Tunlan coalmine." The idea that pore pressure increases create an decrease in permeability and then an increase is an interesting concept.  I have not seen it before.  I have some more basic questions about the study  that the manuscript did not directly address.  In terms or experimentation and descriptions of what has been presented, more information needs to be added to the manuscript to better asses the results.

Although this is clearly a metallurgical coal, that is about all we know.  We should know more about the rank and physical properties.  Some coal behavior can be related to those properties, proximate, ultimate analysis, etc.  Also coal cleat will be very important to permeability parameters.  Although these are laboratory based tests, cleat orientation will very impactful as will the nature of the cleat, spacing, character (blocky, friable, etc.) presence, and consistency in the samples.

Orientation is also important for the the tested samples.  The stress field orientation, location and position of the samples should be described.  There should be a correlation between stress field and cleat orientation.  I think there are some anthracite coal fields in China that are highly fractured by secondary deformation.  The authors did not identify the actual coal bed being mined or any information about it.

The authors looked at the parameters in the is investigation relative to moisture content.  Different moisture contents are achieved by measuring the as received values and then drying the samples and reintroducing moisture to the samples using the laboratory apparatus.  Nothing about the drying process is included.  Ideally this is done at very low temperatures with appropriate lab systems.  Higher temperatures can modify the coal, making subsequent analysis potentially problematic.

The notion of addressing the effects of moisture in lab a samples on permeability does make sense.  However under in situ conditions would this behavior be observed, would the capacity for transport change these findings?  Perhaps the authors should recognize that potential difference in behavior and cite relevant works in this area.

Reviewer 2 Report

 The material submitted is for research on the combined effects of stress, moisture content and gas pressure on the permeability evolution of coal sample and includes a case study of coking coal sourced from Tunlan Coalmine. This material includes the test results along with their correct analysis of the influence of various factors on the gas permeability of coal samples from a specific deposit. Tests have been carried out in which only one result is the finding that the gas permeability of the coal samples decreases exponentially with increasing moisture content. The similarity index of the presented material, excluding bibliography and citations, is relatively high and amounts to 27%. This similarity mainly involves terminology, which is not plagiarized. However, after analyzing the content of the article, I believe that the content of this article does not correspond to the purpose and scope of the scientific journal "Water". Therefore, the Authors should submit this material to a scientific journal covering research in the field of coal mining and methane extraction from coal mines. Summing up, I believe that this material does not meet the conditions for its publication in the scientific journal "Water"

Reviewer 3 Report

It is very interesting paper. It shows research results in the important today field of coalbed methane production. Some improvement is necessary particularly related to the English phrasing.

Things that should be revised are as follows.

  1. What is “thar” ?
  2. Equation 2 uses axial stress and confining stress, but on charts you show confining stress, please indicate what is axial stress and comment its influence.
  3. Table 1and 2. – It is not necessary to repeat values from previous column as “Fitting parameters”
  4. Conclusion: “The effect of confining stress on the permeability of gas-contained coal samples would be greater than that of axial stress.” is not correct. Axial and confining stress refers to apparatus but you can put sample drilled in any direction.
  5. Finally the most important. “Initial gas pressure” was not defined in the paper. “Gas pressure” was not defined in the paper – how do you measure it? Conclusions containing “gas pressure” are questionable. Is Darcy law granted? If not, you do not measure permeability. In such case I suggest to cut data into valid area or perform interpretation of non Darcy flow.

Round 2

Reviewer 2 Report

As already stated in the previous review, the article can be published in the MDPI publishing house after some minor additions. The Authors supplemented the content of the article in accordance with the Reviewers' recommendations. However, I still believe that the substantive scope of the article does not correspond to the scope of the scientific journal "Water". However, I leave the final decision on this matter to the Editors.

Reviewer 3 Report

Thank you for your answer. I suggest to accept paper in present form.